# StEVE: Adaptive Optimization in a Kronecker-Factored Eigenbasis

## Abstract

Adaptive optimization algorithms such as Adam see widespread use in Deep Learning. However, these methods rely on diagonal approximations of the preconditioner, losing much information about the curvature of the loss surface and potentially leading to prolonged training times. We introduce StEVE (Stochastic Eigenbasis-adaptive Variance Estimation), a novel optimization algorithm that estimates lower order moments in the Kronecker-Factored Eigenbasis (KFE). By combining the advantages of Adam over other adaptive methods with the curvature-aware transformations of methods like KFAC and EKFAC, StEVE leverages second-order information while remaining computationally efficient. Our experiments demonstrate that EVE achieves faster convergence both in step-count and in wall-clock time compared to Adam, EKFAC, and KFAC for a variety of deep neural network architectures.

## 1 Introduction

Deep neural networks have shown state-of-the-art performance across a variety of tasks, including computer vision, natural language processing, and speech recognition. Despite their success, training modern models with large parameter counts often requires extensive computational resources and prolonged training times on high-end specialized hardware. This challenge has spurred significant interest in developing more efficient optimization algorithms so as to reduce training time without sacrificing performance.

Stochastic Gradient Descent (SGD) and its variants are the traditional choice of optimization algorithm for training deep neural networks and remain a dominant choice for many model architectures. SGD optimizes the model parameters $\boldsymbol{\theta}$ by computing the gradient of empirical risk (calculated over a mini-batch of training examples) and moving the model parameters by a small step in that direction. Formally, the $t$-th step is $\boldsymbol{\theta}_{t+1} = \boldsymbol{\theta}_t - \eta\nabla_{\boldsymbol{\theta}}\mathcal{R}(\boldsymbol{\theta}_t)$ where $\boldsymbol{\theta}_t$ represents the model parameters at the $t$th step, $\eta$ is a positive learning rate, and $\nabla_{\boldsymbol{\theta}}\mathcal{R}(\boldsymbol{\theta}_t)$ is the gradient of the empirical risk $\mathcal{R}(\boldsymbol{\theta})$.

Despite its simplicity and scalability, SGD struggles with the non-convex and ill-conditioned curvature common to deep neural network loss surfaces. As a typical example, the loss surface may have directions with very different curvatures, and thus the impact of the update in one direction may be much larger than in other directions. This imbalance can raise the number of steps until convergence considerably leading to longer training times.

To correct for these limitations, there have been attempts to design optimization algorithms for deep neural networks which employ second-order information such as the curvature. The general form of these methods is to use an update of the form $\boldsymbol{\theta}_{t+1} = \boldsymbol{\theta}_t - \eta\boldsymbol{P}^{-1}\nabla_{\boldsymbol{\theta}}\mathcal{R}(\boldsymbol{\theta}_t)$ where $\boldsymbol{P}$, referred to as the preconditioner, is some matrix that captures local curvature or similar information about the loss surface such as the Hessian used in Newton-Raphson, the Fisher Information Matrix as used in Natural Gradient Descent (Amari, 1998), Generalized Gauss Newton Matrices, or closely related matrices.

The problem with this form of update is that modern deep neural networks have millions or billions of parameters. Thus, while these methods require fewer updates to train, this advantage is overshadowed by the enormous cost of storing and inverting a fully maintained preconditioner which scale quadratically and cubically respectively with the number of parameters. To overcome these issues it becomes necessary to approximate the preconditioner in a way that allows for faster inversion.

By far the most common approximation is to take the preconditioner to be diagonal. This reduces inversion to pure element-wise computations and also greatly reduces storage cost. Several popular optimization algorithms use this strategy in some form.

1. Adagrad (Duchi et al., 2011) keeps a simple moving average of the elementwise squares of the gradients and elementwise scales the gradients by the inverse square root of this average. In essence, this approach is using a diagonal approximation of the square root of the empirical Fisher

2. RMSProp (Tieleman & Hinton, 2012) uses a similar strategy but uses an exponential moving average of squared gradients.

3. Adam (Kingma & Ba, 2015) introduces bias correction on the exponential moving average and use a different moving average for the gradients themselves.

While these methods have been shown to be more effective in a variety of tasks (Savarese et al., 2021), they only capture curvature information along parameter axes and ignore interactions between different parameters. Consequently, these methods lose much of the second-order information and do not fully correct for poor curvature in the loss surface.

More sophisticated methods avoid diagonal approximations and instead approximate the preconditioner in ways that account for parameter correlations as encoded in the non-diagonal entries of the preconditioner. These approaches vary, although common themes include low rank updates to the preconditioner (Ollivier, 2015; 2017; Mu et al., 2022), using block approximations of the preconditioner or of its inverse (Martens & Grosse, 2015; Desjardins et al., 2015; Fujimoto & Ohira, 2018; Soori et al., 2022), quasi-Newton methods to estimate either the entire preconditioner or its block approximations (Liu & Nocedal, 1989; Goldfarb et al., 2020) and Bayesian inverse-free approaches (Lin et al., 2023; 2024).

Perhaps the most common non-diagonal concept for use in second-order optimization algorithms for deep learning is Kronecker-Factored Approximate Curvature (KFAC). Originally developed for fully-connected layers in Martens & Grosse (2015), KFAC approximates the preconditioning matrix as block diagonal with blocks for each layer and then further approximates each block as a Kronecker product of two smaller matrices. Since inversion commutes with the Kronecker product, this allows for a faster computation of the inverse for each update. This approach has been expanded to convolutional layers in Grosse & Martens (2016) and to weight-sharing layers in Eschenhagen et al. (2023).

Of particular interest is a further refinement of KFAC, Eigenvalue-corrected Kronecker Factored Approximate Curvature (EKFAC) George et al. (2018), which more accurately captures the curvature in different directions by correcting the eigenvalues in KFAC. This is done by diagonalizing the Kronecker factors of the preconditioner blocks and replacing the diagonal with variances in the Kronecker-Factored Eigenbasis (KFE). Due to the expensive nature of the computing KFE, EKFAC amortizes this computation by updating it infrequently while still being able to compute cheap updates to the diagonal variances every iteration. Despite its advantages, EKFAC, even when augmented with momentum, still underperforms Adam in convergence speed for some tasks.

Motivated by the strength of Adam within the scope of diagonal approximations and the curvature-aware properties of EKFAC, we propose STEVE (Stochastic Eigenbasis-adaptive Variance Estimation) which combines the moment estimation of Adam with the curvature corrections of EKFAC. Similar to EKFAC, STEVE transforms the gradients into the KFE but instead of keeping a simple average of second moments STEVE keeps bias-corrected exponential moving averages of the first and second moment in the same way as is done in Adam.

## 2 BACKGROUND AND NOTATION

We consider the supervised learning setup with a training set $\mathcal{D}_{\text{train}}$ consisting of input-output examples $(\boldsymbol{x}, \boldsymbol{y})$ and neural network parametrized by $\boldsymbol{\theta} \in \mathbb{R}^{n_{\theta}}$ which computes a function $f_{\boldsymbol{\theta}}(\boldsymbol{x})$. Our task is to find a value of $\boldsymbol{\theta}$ which minimizes empirical risk $\mathcal{R}(\boldsymbol{\theta}) = \mathbb{E}_{(\boldsymbol{x}, \boldsymbol{y}) \in \mathcal{D}_{\text{train}}}[\mathcal{L}(\boldsymbol{y}, f_{\boldsymbol{\theta}}(\boldsymbol{x}))]$ where $\mathcal{L}$ is some loss function that measures the accuracy of the predictions. Usually, our loss function (e.g. with cross-entropy loss or with MSE loss) can be expressed as negative log probability of a simple

predictive distribution $R_{\boldsymbol{y}|\boldsymbol{z}}$, with density $r(\boldsymbol{y}|\boldsymbol{z})$, parametrized by our neural networks output $\boldsymbol{z}$: $\mathcal{L}(\boldsymbol{y}, \boldsymbol{z}) = -\log r(\boldsymbol{y}|\boldsymbol{z})$. In this context, letting $P_{\boldsymbol{y}|\boldsymbol{x}}(\boldsymbol{\theta}) = R_{\boldsymbol{y}|f_{\boldsymbol{\theta}}(\boldsymbol{x})}$ be the conditional distribution defined by our neural network with density function $p(\boldsymbol{y}|\boldsymbol{x}, \boldsymbol{\theta}) = r(\boldsymbol{y}|f_{\boldsymbol{\theta}}(\boldsymbol{x}))$ we view minimization of empirical risk as maximum likelihood learning of $P_{\boldsymbol{y}|\boldsymbol{x}}$.

We consider algorithms which use stochastic gradients $\nabla_{\boldsymbol{\theta}} = \nabla_{\boldsymbol{\theta}} \mathcal{R}(\boldsymbol{y}, f_{\boldsymbol{\theta}}(\boldsymbol{x})) = \left( \frac{\partial \mathcal{R}(\boldsymbol{y}, f_{\boldsymbol{\theta}}(\boldsymbol{x}))}{\partial \boldsymbol{\theta}} \right)^T$ or averages of them over a mini-batch $\mathcal{B} \subset \mathcal{D}_{\text{train}}$ as computed via backpropagation. Stochastic Gradient Descent updates $\boldsymbol{\theta}_{t+1} = \boldsymbol{\theta}_t - \eta \nabla_{\boldsymbol{\theta}}$ where $\eta$ is a small positive learning rate. Second order methods use a preconditioner $\boldsymbol{A}$ and update as $\boldsymbol{\theta}_{t+1} = \boldsymbol{\theta}_t - \eta \boldsymbol{A}^{-1} \nabla_{\boldsymbol{\theta}}$. Natural Gradient Descent (Amari, 1998) takes $\boldsymbol{A}$ to be the Fisher Information Matrix which, in the case of negative log probability losses, can be expressed as $\boldsymbol{F} = \mathbb{E}_{\boldsymbol{x} \sim \mathcal{D}_{\text{train}}, \boldsymbol{y} \sim p(\boldsymbol{y}|\boldsymbol{x}, \boldsymbol{\theta})}[\nabla_{\boldsymbol{\theta}} \nabla_{\boldsymbol{\theta}}^T]$ where $\boldsymbol{y}$ is sampled from the conditional probability defined by the model. The use of the Fisher as a preconditioner is motivated in Information Geometry as giving the direction of steepest descent in the space of realizable distributions where the metric locally approximates the square root of the KL divergence (Amari & Nagaoka, 2007; Martens, 2020). We use a common approximation of the Fisher which replaces the samples with the labels $\boldsymbol{y}$ from the training set and so we instead have $\boldsymbol{A} = \mathbb{E}_{\boldsymbol{x}, \boldsymbol{y} \sim \mathcal{D}_{\text{train}}}[\nabla_{\boldsymbol{\theta}} \nabla_{\boldsymbol{\theta}}^T]$. The degree to which the Empirical Fisher accurately approximates the Fisher is not clear (Kunstner et al., 2019), but this implementation lowers cost, simplifies implementation and has performed well in practice. Additionally, viewing training from the Langevin Dynamics perspective of gradient flow, preconditioning by the Empirical Fisher gives a stationary Gibbs distribution which is of importance in the realm of statistical mechanics where Langevin Dynamics originates(McAllester, 2023).

Due to its immense size of $n_{\boldsymbol{\theta}} \times n_{\boldsymbol{\theta}}$, inverting and storing $\boldsymbol{A}$ directly is impractical and so we must make a series of approximations. The simplest approximation is to ignore cross-parameter terms entirely and take $\boldsymbol{A}$ to be diagonal. While crude, this comes at an immense advantage in the computational cost of each step. Many optimization algorithms have used variations of this approximation. While these methods seemingly only differ slightly, the impact of these modifications can be substantial. Perhaps the most common such method for use in Deep Neural Networks is Adam (Kingma & Ba, 2015) which keeps track of a bias-corrected exponential moving average of the first moment $\boldsymbol{m}$ and second moment $\boldsymbol{v}$ and updates as follows:

$$\boldsymbol{m}_{t+1} = \beta_1 \boldsymbol{m}_t + (1 - \beta_1) \nabla_{\boldsymbol{\theta}}(\boldsymbol{\theta}_t) \quad \boldsymbol{v}_{t+1} = \beta_2 \boldsymbol{v}_t + (1 - \beta_2) \nabla_{\boldsymbol{\theta}}(\boldsymbol{\theta}_t) \odot$$

$$\hat{\boldsymbol{m}}_{t+1} = \frac{\boldsymbol{m}_{t+1}}{1 - \beta_1^{t+1}} \quad \hat{\boldsymbol{v}}_{t+1} = \frac{\boldsymbol{v}_{t+1}}{1 - \beta_2^{t+1}}$$

$$\boldsymbol{\theta}_{t+1} = \boldsymbol{\theta}_t - \eta \frac{\hat{\boldsymbol{m}}_{t+1}}{\sqrt{\hat{\boldsymbol{v}}_{t+1}} + \epsilon}$$

where squaring, square-rooting, vector-multiplication of $\epsilon$ are done element-wise, $\beta_1, \beta_2$ are hyperparameters for weighing the exponential moving averages, $\hat{\boldsymbol{m}}$ and $\hat{\boldsymbol{v}}$ give the bias corrected first and second moments, and $\epsilon$ is a damping parameter used for numerical stability of inverting the second moment.

Turning now to more elaborate approximations of the preconditioner, most methods exploit the layered structure of Neural Networks and ignore cross-layer terms. Mathematically, if we have $L$ layers this means taking $\boldsymbol{A}$ to be block diagonal:

$$\boldsymbol{A} \approx \bigoplus_{l=1}^{L} \boldsymbol{A}^{(l)}$$

with each block $\boldsymbol{A}^{(l)}$ accounting for the parameters in the $l$th layer. In particular if $\boldsymbol{\theta}^{(l)}$ are the parameters for the $l$th layer, we have $\boldsymbol{A}^{(l)} = \mathbb{E}[\nabla_{\boldsymbol{\theta}^{(l)}} \nabla_{\boldsymbol{\theta}^{(l)}}^T]$ (and the expectation is taking according to the corresponding distribution for either Fisher or Empirical Fisher).

Unfortunately, large layers can still have enough parameters that these blocks can still be too large to invert and store. One solution to this problem, proposed in Martens & Grosse (2015), is to approximate $\boldsymbol{A}^{(l)} \approx \boldsymbol{B}^{(l)} \otimes \boldsymbol{C}^{(l)}$ where $\otimes$ is the Kronecker Product defined as follows:

$$\boldsymbol{V} \otimes \boldsymbol{U} = \begin{bmatrix} V_{1,1}\boldsymbol{U} & V_{1,2}\boldsymbol{U} & \cdots \\ V_{2,1}\boldsymbol{U} & V_{2,2}\boldsymbol{U} & \cdots \\ \vdots & \vdots & \ddots \end{bmatrix}$$

The Kronecker product has many nice algebraic properties which cheapen the cost of updates when used to approximate the preconditioner. For invertible $\boldsymbol{B}, \boldsymbol{C}$, we have $(\boldsymbol{B} \otimes \boldsymbol{C})^{-1} = \boldsymbol{B}^{-1} \otimes \boldsymbol{C}^{-1}$. Thus, if the Kronecker factors have size $a, b$ this reduces cost of inversion from $O((a + b)^3) = O(a^3 + 3a^2b + 3ab^2 + b^3)$ to $O(a^3 + b^3)$ and the cost of storage from $O((a+b)^2) = O(a^2 + 2ab + b^2)$ to $O(a^2 + b^2)$. Similarly, letting vec be the operation which flattens a matrix into a column vector by stacking all of its columns together, we have $\boldsymbol{B} \otimes \boldsymbol{C}\mathrm{vec}(\boldsymbol{D}) = \boldsymbol{C}^T \mathrm{vec}(\boldsymbol{D})\boldsymbol{B}$ reduces the complexity of multiplying preconditioning matrix by gradient.

Specifically, consider a fully connected layer $l$ with input $\boldsymbol{h}$ and pre-activation output

$$\boldsymbol{a} = \boldsymbol{W}\bar{\boldsymbol{h}}$$

where we write the input in homogenous coordinates $\bar{\boldsymbol{h}} = [\boldsymbol{h}, 1]^T$. Then, if $\boldsymbol{g} = \nabla_{\boldsymbol{a}}\mathcal{R}$ is the backpropagated gradient, we have that

$$\nabla_{\boldsymbol{W}} = \boldsymbol{g}\bar{\boldsymbol{h}}^T$$

and thus

$$\nabla_{\boldsymbol{\theta}^l} = \mathrm{vec}(\nabla_{\boldsymbol{W}}) = \bar{\boldsymbol{h}} \otimes \boldsymbol{g}$$

Since $\boldsymbol{A}^{(l)} = \mathbb{E}[\nabla_{\boldsymbol{\theta}^{(l)}}\nabla_{\boldsymbol{\theta}^{(l)}}^T]$, substituting we get the following expression for the Fisher Block

$$\boldsymbol{A}^{(l)} = \mathbb{E}[(\bar{\boldsymbol{h}} \otimes \boldsymbol{g})(\bar{\boldsymbol{h}} \otimes \boldsymbol{g})^T] = \mathbb{E}[(\bar{\boldsymbol{h}}\bar{\boldsymbol{h}}^T) \otimes (\boldsymbol{g}\boldsymbol{g}^T)]$$

We then approximate: $\mathbb{E}[(\bar{\boldsymbol{h}}\bar{\boldsymbol{h}}^T) \otimes (\boldsymbol{g}\boldsymbol{g}^T)] \approx \mathbb{E}[\bar{\boldsymbol{h}}\bar{\boldsymbol{h}}^T] \otimes \mathbb{E}[\boldsymbol{g}\boldsymbol{g}^T]$ which give us our $\boldsymbol{B}^{(l)}$ and $\boldsymbol{C}^{(l)}$.

A very similar principle has been used to extend the KFAC approximation to convolutional layers in Grosse & Martens (2016) and to weight sharing layers in (Eschenhagen et al., 2023).

An instructive perspective on the diagonal approximation of the preconditioner is to view the preconditioner as a diagonal rescaling of the parameter axis as viewed in the parameter basis. Natural Gradient Descent which uses the Fisher $\boldsymbol{A}$ as a preconditioner can also be viewed as a diagonal rescaling. If we diagonalize the positive semi-definite $\boldsymbol{A}$ as $\boldsymbol{A} = USU^T$, the update becomes $\boldsymbol{\theta}_{t+1} = \boldsymbol{\theta}_t - US^{-1}U^T\nabla_{\boldsymbol{\theta}}\mathcal{L}$ which is to say converting the gradient $\boldsymbol{A}$'s Eigenbasis, doing a diagonal rescaling by the eigenvalues of the Fisher, and then switching back to the parameter basis. This perspective poses a challenge to the KFAC approximation as the critically important eigenvalues of the Fisher Blocks are not preserved by the approximation.

EKFAC (George et al., 2018) addresses this issue by correcting the eigenvalues of the KFAC approximation. They do this by diagonalizing $\boldsymbol{A}^{(l)} = \boldsymbol{B}^{(l)} \otimes \boldsymbol{C}^{(l)} = (\boldsymbol{U_B} \otimes \boldsymbol{U_C})(\boldsymbol{S_B} \otimes \boldsymbol{S_C})(\boldsymbol{U_B} \otimes \boldsymbol{U_C})^T$ and then replacing $(\boldsymbol{S_B} \otimes \boldsymbol{S_C})$ with $\mathrm{diag}(\mathbb{E}[(\boldsymbol{U_B} \otimes \boldsymbol{U_C})^T\nabla_{\boldsymbol{\theta}}\mathcal{R}^2])$ which is the matrix with diagonal equal to the vector of second moments in Kronecker-Factored Eigenbasis (KFE) defined by applying the transformation $(\boldsymbol{U_B} \otimes \boldsymbol{U_C})^T$. This replacement yields a provably closer approximation to the Fisher (as measured by the Froebenius Norm) and the optimal diagonal scaling in the KFE. Additionally, this approximation lends itself well to amortizing the expensive curvature estimation as the KFE does not have to updated with every step while the diagonal matrix of eigenvalues can cheaply be updated every step. Unfortunately, even when augemented with running averages EKFAC struggles to compete with Adam in practice.

## 3 PROPOSED METHOD

Our proposed method, STEVE, builds upon the insights from EKFAC and the success of Adam in the realm of diagonal adaptive optimizers. Viewing EKFAC from the perspective of diagonal

rescaling, it effectively rescales the gradients by the second moments computed in the KFE. This observation suggests that we can apply other diagonal adaptive optimization methods in the KFE.

In particular, we propose leveraging the advancements of Adam within the KFE framework. STEVE operates similarly to EKFAC in that it periodically computes the KFE for each Fisher block. However, instead of using only the second moments, STEVE maintains bias-corrected exponential moving averages of both the first and second moments of the gradients in the KFE, estimated in the same manner as in Adam. By combining the benefits of the Kronecker-factored approximation with the adaptive moment estimation of Adam, STEVE aims to achieve faster convergence.

---

**Algorithm 1** STEVE

---

**Require:** $n$: Recompute KFE every $n$ minibatches
**Require:** $\eta$: Learning rate
**Require:** $\beta_1$: Momentum parameter for first moment
**Require:** $\beta_2$: Momentum parameter for second moment
**Require:** $\epsilon$: Damping parameter
1: **procedure** STEVE(Train)
2:     **while** convergence is not reached, iteration $i$ **do**
3:         Sample minibatch $\mathcal{B}$ from Train
4:         Forward pass to obtain $\bar{h}$ and backprop to obtain $g$
5:         **for all** layer $l$ **do**
6:             $c \leftarrow i \bmod n$
7:             **if** $c = 0$ **then**
8:                 COMPUTE-KFE($\mathcal{B}, l$)
9:             **end if**
10:            COMPUTE-SCALINGS($\mathcal{B}, l$)
11:            UPDATE-PARAMETERS($\mathcal{B}, l$)
12:         **end for**
13:     **end while**
14: **end procedure**
15: **procedure** COMPUTE-KFE($\mathcal{B}, l$)
16:     $U_B^{(l)}, S_B^{(l)} \leftarrow$ eigendecomposition $\left(\mathbb{E}_{\mathcal{B}}[h^{(l)}h^{(l)T}]\right)$
17:     $U_C^{(l)}, S_C^{(l)} \leftarrow$ eigendecomposition $\left(\mathbb{E}_{T}[g^{(l)}g^{(l)T}]\right)$
18:     $m, v \leftarrow 0$
19: **end procedure**
20: **procedure** COMPUTE-SCALINGS($\mathcal{B}, l$)
21:     $m \leftarrow \beta_1 m + (1 - \beta_1)\mathbb{E}_{\mathcal{B}}\left[\left(U_B^{(l)} \otimes U_C^{(l)}\right)^T \nabla_\theta^{(l)}\right]$

22:     $v \leftarrow \beta_2 v + (1 - \beta_2)\mathbb{E}_{\mathcal{B}}\left[\left(\left(U_B^{(l)} \otimes U_C^{(l)}\right)^T \nabla_\theta^{(l)}\right)^2\right]$

23: **end procedure**
24: **procedure** UPDATE-PARAMETERS($\mathcal{B}, l$)
25:     $\hat{m} = \frac{m}{\sqrt{1-\beta_1^c}}$
26:     $\hat{v} = \frac{v}{\sqrt{1-\beta_2^c}}$
27:     $\tilde{\nabla} \leftarrow \frac{\hat{m}}{\sqrt{\hat{v}} + \epsilon}$
28:     $\nabla_F \leftarrow \left(U_B^{(l)} \otimes U_C^{(l)}\right)\tilde{\nabla}$
29:     $\theta^{(l)} \leftarrow \theta^{(l)} - \eta\nabla_F$
30: **end procedure**

---

## 4 EMPIRICAL RESULTS

In this section, we present empirical evaluations of STEVE across a variety of datasets and model architectures. All experiments were conducted on a single NVIDIA A100 through Google Colab us-

ing PyTorch (Paszke et al., 2017). We compare against against Adam, EKFAC, and KFAC showing favorable comparisons for STEVE in terms of both Epoch Count and Wall-Clock Time. For classification tasks, we train the model on a constant learning rate until the model reaches a test accuracy past a pre-determined cutoff consistent with what the model usually reaches after approximately 100 epochs on Adam. We rely on the implementation of KFAC for convolutional layers (Grosse & Martens, 2016) and the implementation of KFAC-reduce for Attention layers (Eschenhagen et al., 2023). All optimizers except Adam are implemented as preconditioners on top of SGD.

## 4.1 RESNET-50 ON CIFAR-10

To evaluate the effectiveness of STEVE, we first conducted experiments on the CIFAR-10 dataset using a ResNet-50 architecture. We compared STEVE with Adam, EKFAC, and KFAC, training each model until it reached a test accuracy of $92.5\%$. All optimizers used a constant learning rate of 0.001. EKFAC and KFAC employed running averages to estimate curvature, updating their curvature estimates every 500 steps; STEVE followed the same schedule. For Adam and STEVE, we set the hyperparameters to $\beta_1 = 0.9$, $\beta_2 = 0.999$, and $\epsilon = 10^{-8}$, while EKFAC and KFAC used $\alpha = 0.9$. Each model was allowed to train for a maximum of 100 epochs. Data preprocessing included random cropping and horizontal flipping for the training data, and normalization for both training and test sets.

Figure 1 displays the performance of the different optimizers over wall-clock time and epochs. Notably, STEVE achieved the target accuracy significantly faster than the other methods. Specifically, STEVE demonstrated a **40% reduction in wall-clock time** and a **60% reduction in the number of epochs** compared to Adam. The other methods did not converge at this learning rate.

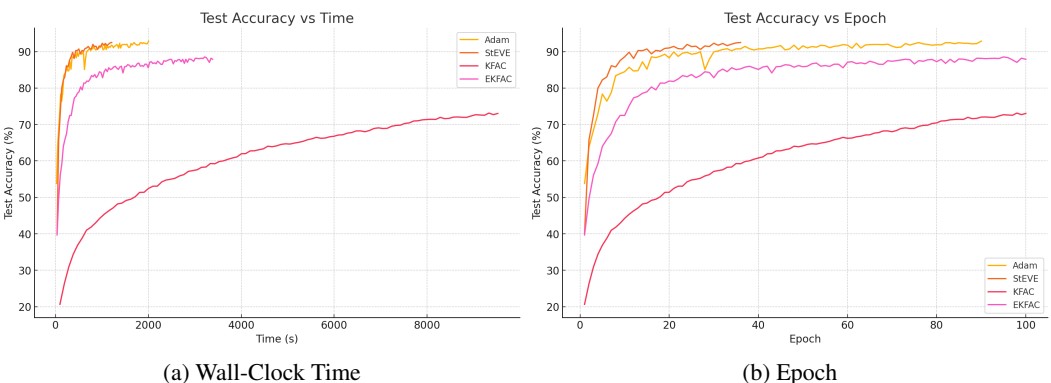

(a) Wall-Clock Time       (b) Epoch

Figure 1: CIFAR-10 ResNet-50. **(a)** Test loss vs wall-clock time. **(b)** Training loss vs Epoch.

## 4.2 RESNET-50 ON TINY IMAGENET

We further assess the performance of STEVE on the more challenging Tiny ImageNet dataset, again utilizing a ResNet-50 architecture. We compared STEVE against Adam, EKFAC, and KFAC, training until the models reached a test accuracy of 44%. A learning rate of 0.0001 was used across all optimizers. Similar to the previous experiment, EKFAC and KFAC used running averages for curvature estimation, updating every 600 steps, with STEVE following the same schedule. Hyperparameters for Adam and STEVE were set to $\beta_1 = 0.9$, $\beta_2 = 0.999$, and $\epsilon = 10^{-8}$, while EKFAC and KFAC used $\alpha = 0.9$. Training was capped at 100 epochs. The data preprocessing pipeline included random cropping and horizontal flipping for the training data, along with normalization for both training and test sets.

As illustrated in Figure 2, STEVE outperformed the other optimizers by a substantial margin. It achieved the target accuracy with a **60% reduction in wall-clock time** and an **85% reduction in the number of epochs** compared to Adam. Once again, EKFAC and KFAC failed to converge within the allocated epochs, underscoring the effectiveness of STEVE in handling more complex datasets.

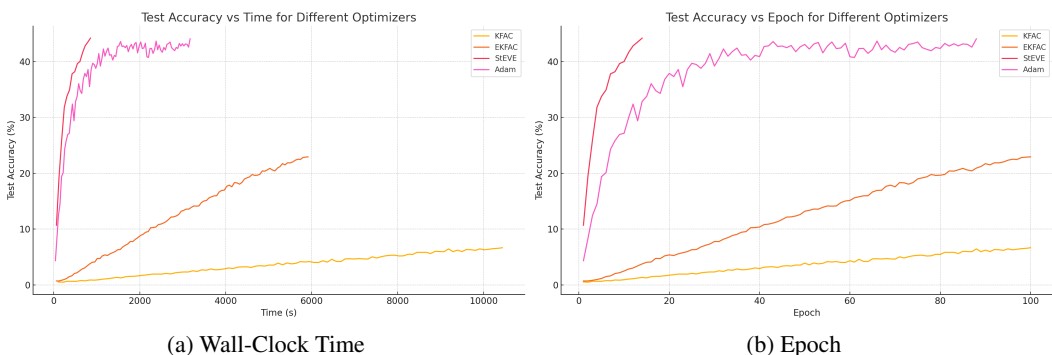

(a) Wall-Clock Time          (b) Epoch

Figure 2: Tiny ImageNet ResNet-50. **(a)** Test loss vs wall-clock time. **(b)** Training loss vs Epoch. STEVE shows a gain of approximately $60\%$ in wall-clock time and $85\%$ in number of epochs as compared to Adam and the rest of the optimization algorithms do not converge within the allocated epochs.

### 4.3 VIT-S/16 ON CIFAR-100

Finally, we evaluated STEVE on the CIFAR-100 dataset using a Vision Transformer (ViT-S/16) architecture, comparing it against Adam. Note that following the implementation of KFAC for MultiHead Attention layers in Eschenhagen et al. (2023), we reimplement the MultiHead Attention layer using nn.Linear layers for $Q, K, V$. The models were trained until reaching a test accuracy of 46%. All optimizers used a learning rate of 0.00005. STEVE updated its curvature estimates every 50 steps. Hyperparameters for both Adam and STEVE were set to $\beta_1 = 0.9$, $\beta_2 = 0.999$, and $\epsilon = 10^{-8}$. Training was limited to 100 epochs. Data preprocessing involved resizing images to accommodate the patch size of 16, random cropping, random horizontal flipping for the training data, and normalization for both training and test sets.

Figure 3 presents the performance comparison between STEVE and Adam. STEVE achieved the target accuracy with a **30% reduction in wall-clock time** and a **60% reduction in the number of epochs** compared to Adam. These results highlight STEVE's capability to accelerate training even for transformer-based architectures.

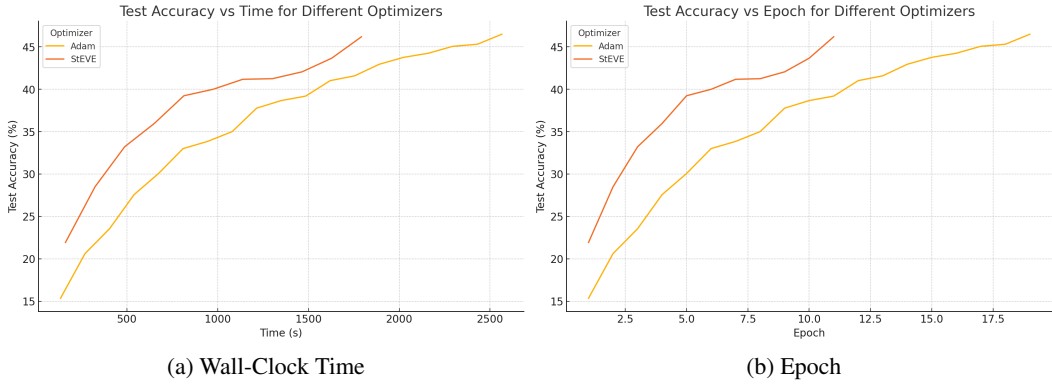

(a) Wall-Clock Time          (b) Epoch

Figure 3: CIFAR-100 ViT-S/16. **(a)** Test loss vs wall-clock time. **(b)** Training loss vs Epoch. STEVE shows a gain of approximately $30\%$ in wall-clock time and $60\%$ in number of epochs as compared to Adam and the rest of the optimization algorithms do not converge within the allocated epochs.

## 5 CONCLUSION AND FUTURE WORK

In this paper, we introduced STEVE, a novel optimization algorithm that synergizes the moment estimation of Adam with the curvature-aware preconditioning of EKFAC. By transforming gradients into a Kronecker-Factored Eigenbasis (KFE) of the Fisher and maintaining bias-corrected exponential moving averages of the first and second moments, STEVE leverages second-order information while retaining computational efficiency. Our empirical evaluations across various datasets and architectures demonstrate that STEVE significantly accelerates training, achieving substantial reductions in both wall-clock time and number of epochs compared to existing optimization algorithms such as Adam, EKFAC, and KFAC.

Despite promising results, there are avenus for future exploration and improvement. One direction to take is to improve the KFE by attempting to use other common preconditioners instead of the Empirical Fisher such as the true Fisher Information Matrix. Other directions to take the work are to investigate the potential of the improvements that have been made over Adam in the KFE such as proper weight decay or Nesterov momentum.

### REPRODUCIBILITY STATEMENT

We are committed to the reproducibility of our results and have taken the necessary steps to ensure this. In the supplementary materials, we provide comprehensive code for all preconditioners used in our benchmarks, including implementations of the proposed STEVE optimizer and other optimizers used for benchmarking. The codebase includes the models we trained, detailed data preprocessing steps, and a sample training loop, enabling others to replicate our experiments fully. The Empirical Results section outlines all hyperparameters and training conditions necessary for reproduction. Additionally, our implementation of the optimizer closely follows the pseudocode presented in the Proposed Method section, ensuring transparency and ease of understanding for replication purposes.

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
