# OpenReview forum: "StEVE: Adaptive Optimization in a Kronecker-Factored Eigenbasis"
_ICLR.cc/2025/Conference — ICLR 2025 Conference Withdrawn Submission_

### Official Review · Reviewer_VXLg · 2024-10-29

**Soundness:** 3
**Presentation:** 2
**Contribution:** 3
**Rating:** 3
**Confidence:** 3

**Summary:**

Adamize diagonal corrections to KFAC in a similar way to EKFAC

**Strengths:**

This paper adopts and extends the style of thinking seen in EKFAC to apply diagonal corrections to KFAC.

**Weaknesses:**

The paper opts to use SGD as the base opt for KFAC and EKFAC. The official and unofficial codebases for KFAC allow and some actual suggest to use Adam as the base opt for KFAC and EKFAC. This is because it's well known that when we operate using Adam as the base opt that drives E/KFAC it works better.

The authors say:

"However, instead of using only the second moments, STEVE maintains bias-corrected exponential moving averages of both the first and second moments of the gradients in the KFE, estimated in the same manner as in Adam. By combining the benefits of the Kronecker-factored approximation with the
adaptive moment estimation of Adam, STEVE aims to achieve faster convergence."

While the opt in this pape is not exactly Adam as the base opt driving E/KFAC it is in similar vain, as such it would have been helpful to have ran experiments with SGD and Adam as the base opts for E/KFAC so we could see if there is a delta.

Another weakness is not comparing to Shampoo which is an alternative kronecker factorized optimizer that has become quite popular recently due to its strong performance at Google. Furthermore the same way this paper proposes Adamized diagonal 1st and 2nd moment corrections to KFAC, SOAP proposes this for Shampoo. As such this paper should really compare to those methods.

Furthermore, PSGD Affine or Kronecker factorized has been shown to outperform E/KFAC as well as Shampoo/SOAP and should be compared as well for this paper to be complete.

Another weakness is the use of a ViT for cifar datasets. The images are too small for patches to make sense and so it generally doesn't do well. Something like Keller's modded-nanoGPT would be a good place to show the performance of the opt since it's been benchmarked against all the latest curvature informed optimizers.

**Questions:**

What is the memory and computational complexity of the proposed opt?

How frequently is the preconditioner updated? Shampoo updates every 100 iterations, PSGD updates every 10 iters. It would be good to see how often the precond must be updated and how it effects performance.

Variance bars? A

The claim that the proposed opt significantly outperforms (40% reduction in all clock time) Adam in fig 1 seems not true based on wall clock time. It seems at the end of training Adam ends at a higher accuracy, and Adam actually matches StEVE only a few hundred seconds later. Since the authors do not show variance bars we have no way of knowing if this is a legit speedup.

Furthermore, with the extra memory needed to train with StEVE one could easily boost batch size for Adam and see an improvement in performance.

---

### Official Review · Reviewer_5iqx · 2024-11-04

**Soundness:** 3
**Presentation:** 2
**Contribution:** 4
**Rating:** 8
**Confidence:** 4

**Summary:**

The work proposes STEVE, a novel optimization method that combines the strengths of the Adam optimizer (cheap tracking and adaptation to diagonal second order properties) and EKFAC (amortized better approximation of full second order). This is achieved by applying Adam, not in original parameter space, but in the Kronecker-Factored Eigenbasis (KFE) i.e. the “preconditioning” basis used by KFAC and EKFAC. Experiments on image classification tasks with ResNet-50 and ViT architectures show significantly faster optimization compared to Adam, both in number of epochs and wall-clock time.

**Strengths:**

- Originality: The optimizer developed in the paper is novel: an original combination of the strengths of EKFAC and Adam.

- Significance: This is a significant and timely contribution in the context of a heightened interest in more efficient optimization methods for deep learning (e.g. [1,3]) . The development of better off-the-shelf optimizers suitable for training deep learning models is an essential component for driving progress in the field, as can be seen in the wide adoption of Adam. In spite of their theoretical superiority, non-diagonal second order methods have struggled to manifest practical superiority for training standard deep learning models over their simpler diagonal counterparts. That the proposed method manages to convincingly beat Adam on deep network training tasks, both in number of epochs and in wallclock time, is thus significant. It showcases the potential of the approach and warrants the attention of the community.

- Clarity: Motivation, background, and the proposed method are clearly explained (except for minor glitches, see below). This is in part thanks to a clear algorithm box. I also appreciate that readily usable pytorch code is given in the supplementary for reproducibility. Experimental setup and methodology are also briefly but clearly explained.

- Quality: The approach is well-motivated, appears sound and well implemented, and the presented experiments convincingly support the claim of superiority of the developed optimizer.

**Weaknesses:**

- Missing a more thorough recent related works discussion.
Related work pertaining to background is well covered, but the paper is missing a section discussing later advances in second order optimization methods for deep learning. See [1,2,3] and first question below for starting pointers.

- The experimental analysis could have been pushed further: to include also training loss curves, and an evaluation and discussion of the relative sensitivity to hyperparameters. (see questions section for details).

- Somewhat limited scope and scale of experimental evaluation.
While I value the experimentation on 2 different deep architectures ResNet50 and ViT and 2 image datasets, a more extensive experimentation on a larger variety of tasks and datasets would help to more solidly establish the advantage of the approach. See e.g. deep net training benchmark [1].

- Paper would benefit from a little more polishing.
Some (minor and easily fixable) clarity issues. See questions part for a list and suggested improvements.

**Questions:**

Q1:  I would like to draw your attention to concurrent work SOAP [3], which seems closely related as it also uses Adam inside a second order preconditioning approach, Shampoo [2]. This doesn’t lower the originality of your proposal, being concurrent work that you likely couldn’t know about at the time of submission. But given the relatedness of the approaches, I am interested to know how you would contrast them? What can you highlight as the differences / anticipated benefits & limitations of STEVE=EKFAC+Adam v.s. SOAP=Shampoo+Adam ?
Also, how do they compare in memory and compute complexity?
This discussion could become part of a fleshed out related works section.

Q2:  Algo lines 16 and 17 eigendecomposition(...): what are the expectations over B and T? Can you provide more details on how these expectations are computed/estimated/tracked? (I suggest to also update the algo box to provide this additional level of detail, as well as main text l 283 “running averages”)

Q3: Training loss curves associated with your test accuracy curves.
Can you include these (in supplementary if space is insufficient in main)
Do the higher test accuracies also correspond to lower training losses? Please discuss.

Q4: What are the test accuracy and training loss reached by all algos at the max number of iterations you used?

Q5: Sensitivity to hyperparameters?
Do you have evidence that your optimizer outperforming Adam does not require extensive fine-tuning of (additional?) hyper-parameters. E.g. how sensitive is it to recompute frequency?
Similarly you write l291 “The other methods did not converge at this learning rate”, but would thay at other rates?

Further clarifying suggestions:
- L 200 “as the critically important eigenvalues … are not preserved by the approximation” -> needs more explanation.
- The explanation of EKFAC and in particular the KFE in paragraph line 202 is too dense. This is the algo that you build on, so please try to lighten expand and clarify.
- BUG towards end of update equation for Adam’s  $v_{t+1}$ line 139, missing a square?
- Curves: please use more easily distinguishable colors than different shades of red! (given the chance, make them color-blind friendly, see e.g. https://davidmathlogic.com/colorblind, and/or use different line styles)
- Figure 3 is missing KFAC and EKFAC.


Typos and English fixes:
- Abstract L19: “EVE” -> “STEVE”
- L 148: “vector-multiplication of $\epsilon$ are done element-wise”. I see no vector multiplication of $\epsilon$ ???
- L 161: “is taking” -> “is taken”
- L 175: “reduces” -> “which reduces”
- L 198: “converting” -> “changing to”
- L 270: “against against”
- L 283: “running averages”, computed how exactly?
- L 285: $\alpha$ has never been defined. You should at least say what it is and does in KFAC/EKFAC.
- L 291: “The other methods did not converge at this learning rate” -> do you mean they did to reach the target accuracy? What about at other learning rates? Did you hyper-optimize over it, and how sensitive are the methods to it?
- L 382: “of the Fisher” -> “of the empirical Fisher”
- L 391: “Other directions to take the work are to investigate the potential of the improvements that have been made over Adam in the KFE such as proper weight decay or Nesterov momentum.” -> “Future work should also investigate the potential of using, in the KFE, other improvements that have been made over Adam, such as proper weight decay [ADD REFERENCE] or Nesterov momentum [ADD REFERENCES].



[1] Benchmarking Neural Network Training Algorithms, Dahl et al. 2023 https://arxiv.org/abs/2306.07179

[2] Shampoo: Preconditioned stochastic tensor optimization. V. Gupta, T. Koren, Y. Singer. ICML 2018

[3] SOAP: Improving and Stabilizing Shampoo using Adam. Vyas et al. September 2024.
https://arxiv.org/abs/2409.11321

---

### Official Review · Reviewer_Sser · 2024-11-05

**Soundness:** 1
**Presentation:** 3
**Contribution:** 2
**Rating:** 3
**Confidence:** 3

**Summary:**

This paper presents an optimization method for deep learning, which performs Adam-style adaptation in a Kronecker-factored eigenbasis. The proposed method is evaluated empirically against vanilla Adam as well as other Kronecker-factored optimizers.

**Strengths:**

- The paper gives a good introduction to relevant prior work and the contributions are adequately positioned in the context of prior work.
- The idea for the method is well-motivated, lifting the adaptive Adam scheme to a Kronecker-factored eigenbasis. To my knowledge, this idea has not been explored before and is original.
- The method shows promising initial results in the experimental framework of the paper.
- The paper is generally well-written and easy to follow.

**Weaknesses:**

The proposed method is a straight-forward combination of existing ideas. No supporting theory is provided. In my opinion, such a paper needs a very detailed and fair experimental comparison to warrant publication at ICLR. Unfortunately, the quality of the experiments is subpar. To mention just a few issues I see
- Experiments are run with a single random seed.
- All methods use the same learning rate and it is not explained where that learning rate value comes from. This is not adequate for an empirical comparison of different optimizers.
- Experiments use a constant learning rate instead of established learning rate decay schedules.

**Questions:**

- How was the learning rate for the experiments chosen?
- Why weren't the learning rates set individually for each competing method?

---

### Official Review · Reviewer_xFQj · 2024-11-06

**Soundness:** 2
**Presentation:** 3
**Contribution:** 2
**Rating:** 3
**Confidence:** 4

**Summary:**

The paper introduces StEVE, a novel deep learning optimizer that combines aspects of Adam and KFAC. Specifically, they modify EKFAC, which corrects the eigenvalues of the KFAC approximation, by adding Adam's bias-corrected first and second moment estimators. The authors show that StEVE achieves faster training to a target performance in both step count and wall-clock time compared to Adam, KFAC, and EKFAC, on three different deep learning problems on CIFAR-10, CIFAR-100, and Tiny ImageNet.

**Strengths:**

- The paper presents a novel and interesting approach to incorporating an Adam-style update into the EKFAC optimizer. The resulting algorithm is clearly described in Algorithm 1 and is rather straightforward to implement. The paper also provides code for the new optimizer (as well as the experiments). It presents an extensive introduction and background section and thus an accessible explanation of the method.
- Faster neural network training is a crucial research topic and any progress in this area is of great interest to the entire deep learning community.
- The paper not only focuses on the number of steps but also considers the - practically much more relevant - wall-clock runtime.

**Weaknesses:**

The empirical evidence for StEVE is too weak to be convincing. As there are now hundreds of deep learning optimizers, the empirical burden of proof of superiority is quite high, especially for optimizers like StEVE who are mostly motivated by their empirical performance. I believe the currently provided experiments don't provide enough evidence to convince people to adopt it in practical applications, for the following reasons:

- Most importantly, the hyperparameter selection seems to be performed in an opaque and potentially unfair way. Apparently, no hyperparameter tuning was performed, e.g., with all optimizers sharing the same learning rate. Yet, the selected learning rate differs between experiments (e.g. 0.001 for CIFAR-10 and 0.00005 for CIFAR-100). How was this chosen? I suspect that these choices work well for StEVE, but not the compared baseline. A more meaningful comparison would be to either tune the hyperparameters for each method on each test problem independently (using the same budget) or use fixed hyperparameters for all methods that are shared across all test problems. The latter would be a "hyperparameter-free" optimization and would require different baselines, e.g. Schedule-Free [1].
- All experiments are done on small problems, with CIFAR-100 being the largest. Also, all are from the same data domain and task, namely image classification.
- No learning rate schedule was used. I don't think a constant schedule is a very practical choice.
- Overall, the baselines seem to be very weak, likely due to inefficient hyperparameter choices (see the first point).
- The target performances seem rather impractical, e.g. only 44% on Tiny ImageNet and 46% on CIFAR-100. This is far from the performance that one can achieve on these datasets (with the used models) and thus not a performance practitioners care about. This is relevant because optimizers that can quickly achieve a low performance can be quite different from optimizers that achieve a more competitive performance quickly.

Without a more rigorous evaluation, I doubt that the method will have a significant impact. I suggest having a look at [2], which describes a protocol for comparing deep learning optimizers. Although running the full benchmark might be too computationally expensive, following some of the described practices could significantly strengthen the empirical evidence for StEVE and thus demonstrate its strength more convincingly.

[1] Aaron Defazio, Xingyu Alice Yang, Harsh Mehta, Konstantin Mishchenko, Ahmed Khaled, Ashok Cutkosky; "The Road Less Scheduled"; arXiv 2024; <https://arxiv.org/abs/2405.15682>

[2] George E. Dahl et al.; "Benchmarking Neural Network Training Algorithms"; arXiv 2023; <https://arxiv.org/abs/2306.07179>

**Questions:**

- Why is KFAC so much slower per step compared to EKFAC? E.g. in Figure 1, both KFAC and EKFAC perform 100 epochs, yet KFAC requires roughly 3x the wall-clock time.
- Could you add a short paragraph providing a complexity analysis of the computational and memory requirements of StEVE compared to Adam and EKFAC? In my understanding, it should be very similar to EKFAC in both time per step and memory, with the additional memory of a second EMA (for the first moment). Is this correct?
- Line 19: Should it be "StEVE" instead of "EVE"?
- Suggestion: Both Section 1 and Section 2 extensively describe existing work. Only at the bottom of page 4, do you start describing your own method. If you compress Sections 1 and 2, you have more space to present your method, which I think would strengthen your paper.
- In the paragraph starting at line 79, I think it might be worth mentioning and discussing Shampoo [e.g. 3] and related methods. Shampoo recently won the AlgoPerf: Training Algorithms competition and seems to be a practically relevant non-diagonal method (with likely use in training Gemini models).
- Line 88: George et al. should probably be a parencite or citep.
- Line 91: It should probably be "Due to the expensive nature of [] computing [the] KFE ".
- Line 127: There should probably be a space before the citation.
- In Adam's equation, I think there is something missing for the EMA of the second momentum. Either a second gradient after the element-wise multiplication or rather a square (since you mention squaring below).
- Also just below the equation (line 148) you mention "vector-multiplication of $\epsilon$. Do you mean "addition"? I don't see where $\epsilon$ is multiplied.
- Is there a reason that Section 1 uses $\mathbf{P}$ as the preconditioner (line 45) and in Section 2 you use $\mathbf{A}$ (line 116) instead?
- Line 197: I think $USU$ should also be bolded, since you use bold-face for matrices, no?
- Line 198: Is this sentence missing a "to", i.e. "which is to say converting the gradient [to] $\mathbf{A}$'s Eigenbasis"?
- In Algorithm 1, you could highlight the differences between StEVE and EKFAC, e.g. by coloring lines that changed.
- Line 270: There is a double "against".
- Line 271: "Epoch Count" and "Wall-Clock Time" should probably both be lowercase.
- The figures, and especially the legends are relatively small and thus hard to read.
- In the figures, try using a consistent coloring/legend. For example, Adam is yellow in Figure 1 but in Figure 2 KFAC is yellow. This makes it hard to quickly compare across figures. The colors are also relatively similar (yellow, orange, red, pink) and thus hard to distinguish.
- Is there a reason to not compare to KFAC and EKFAC for the ViT on CIFAR-100?

[3] Rohan Anil, Vineet Gupta, Tomer Koren, Kevin Regan, Yoram Singer; "Towards Practical Second Order Optimization for Deep Learning"; OpenReview 2021; <https://openreview.net/forum?id=Sc8cY4Jpi3s>

---

### Note · Authors · 2024-11-28

I have read and agree with the venue's withdrawal policy on behalf of myself and my co-authors.